# Citrate-Induced p85α–PTEN Complex Formation Causes G_2_/M Phase Arrest in Human Pharyngeal Squamous Carcinoma Cell Lines

**DOI:** 10.3390/ijms20092105

**Published:** 2019-04-29

**Authors:** Kuang-Chen Hung, Shyang-Guang Wang, Meng-Liang Lin, Shih-Shun Chen

**Affiliations:** 1Division of Neurosurgery, Department of Surgery, Taichung Army Force General Hospital, Taichung 41152, Taiwan; sur060@gmail.com; 2Department of Surgery, National Defense Medical Center, Taipei 11490, Taiwan; 3General Education Center, College of Humanities and General Education, Central Taiwan University of Science and Technology, Taichung 40601, Taiwan; 4Department of Medical Laboratory Science and Biotechnology, Central Taiwan University of Science and Technology, Taichung 40601, Taiwan; sgwang@ctust.edu.tw; 5Department of Medical Laboratory Science and Biotechnology, China Medical University, Taichung 40402, Taiwan

**Keywords:** Akt, citrate, cyclin B1, CDK1, p85α, PTEN, G_2_/M

## Abstract

Citrate is a key intermediate of the tricarboxylic acid cycle and acts as an allosteric signal to regulate the production of cellular ATP. An elevated cytosolic citrate concentration inhibits growth in several types of human cancer cells; however, the underlying mechanism by which citrate induces the growth arrest of cancer cells remains unclear. The results of this study showed that treatment of human pharyngeal squamous carcinoma (PSC) cells with a growth-suppressive concentration of citrate caused cell cycle arrest at the G_2_/M phase. A coimmunoprecipitation study demonstrated that citrate-induced cell cycle arrest in the G_2_/M phase was associated with stabilizing the formation of cyclin B1–phospho (p)-cyclin-dependent kinase 1 (CDK1) (Thr 161) complexes. The citrate-induced increased levels of cyclin B1 and G_2_/M phase arrest were suppressed by the caspase-3 inhibitor Ac-DEVD-CMK and caspase-3 cleavage of mutant p21 (D112N). Ectopic expression of the constitutively active form of protein kinase B (Akt1) could overcome the induction of p21 cleavage, cyclin B1–p-CDK1 (Thr 161) complexes, and G_2_/M phase arrest by citrate. p85α–phosphatase and tensin homolog deleted from chromosome 10 (PTEN) complex-mediated inactivation of Akt was required for citrate-induced G_2_/M phase cell cycle arrest because PTEN short hairpin RNA or a PTEN inhibitor (SF1670) blocked the suppression of Akt Ser 473 phosphorylation and the induction of cyclin B1–p-CDK1 (Thr 161) complexes and G_2_/M phase arrest by citrate. In conclusion, citrate induces G_2_/M phase arrest in PSC cells by inducing the formation of p85α–PTEN complexes to attenuate Akt-mediated signaling, thereby causing the formation of cyclin B1–p-CDK1 (Thr 161) complexes.

## 1. Introduction

Reprogramming of cellular metabolism from respiration toward aerobic glycolysis is recognized to supply the nucleotides, proteins, and lipids needed for rapid cell division, continuous growth, invasion, metastasis, and resistance to chemotherapeutic agents [1,2,3]. Although an elevated level of the aerobic glycolysis-derived pyruvate is preferentially converted into lactic acid, some of the pyruvate is transported to the mitochondrial matrix, where it is diverted into acetyl-coenzyme A (acetyl-CoA) [4]. Mitochondrial acetyl-CoA and oxaloacetate (OAA) undergo a condensation reaction by citrate synthase to form citrate, which is then transported to the cytosol to convert acetyl-CoA by ATP-citrate lyase (ACL). Cytosolic acetyl-CoA serves as a precursor for de novo lipogenesis and is first carboxylated to malonyl-CoA by acetyl-CoA carboxylase (ACC) [4]. ACCs are occasionally found to be overexpressed in cancer cells and are correlated with a poor clinical outcome [5,6,7]. However, high levels of citrate are present in the cytosol that suppress phosphofructokinase, a key enzyme of glycolysis, consequently inhibiting ATP production and the de novo synthesis of phospholipids [8]. Citrate can also inhibit the activity of pyruvate dehydrogenase, resulting in decreased synthesis of ATP and lipids [9,10]. Deprivation of ATP by glycolytic inhibition can induce apoptosis and inhibit cell growth in drug-resistant cancer cells [11,12], indicating that depletion of the cellular ATP supply may be an effective way to overcome chemoresistance in cancer therapy.

Serine/threonine kinase protein kinase B (PKB/Akt)-mediated signaling is critical for proliferation, survival, metabolism, and metastasis of cancer cells [13] and is triggered by class IA phosphatidylinositol 3–kinase (PI3K) induction of the generation of phosphatidylinositol-3,4,5-trisphosphate (PIP_3_) [14]. Although the p85α regulatory subunit is crucial in mediating the activation of PI3K, p110α-free p85α homodimers have the ability to bind and positively regulate the lipid phosphatase activity of phosphatase and tensin homolog deleted from chromosome 10 (PTEN) to modulate Akt-mediated signaling [15]. Akt activity is negatively regulated by PTEN through dephosphorylating the 3-position of PIP_3_ to phosphatidylinositol-4,5-bisphosphate [16,17]. Increased activity and dysregulation of Akt is associated with chemoresistance in various human cancer cells [18]. Inactivation of Akt by chemotherapeutic drugs or the PI3K inhibitor LY294002 induces cell cycle arrest in the G_2_/M phase [19]. The mechanistic ability of Akt signaling to control the survival of cancer cells has been documented to increase the stability of p21 via phosphorylation on p21 [20]. In the presence of p21, overexpression decreases the induction of G_2_/M phase arrest and apoptosis by paclitaxel in p53-null human sarcoma cells [21]. Active Akt exerts its action on the regulation of glucose uptake through the inhibition of Bax conformation change [22]. Previous studies have indicated that Akt enhances the efflux of citrate from the mitochondria to the cytosol in the form of acetyl-CoA by phosphorylating and activating ACL [23,24,25]. Additionally, Akt has been shown to be involved in the stimulation of aerobic glycolysis in cancer cells [26], motivating us to explore whether citrate attenuates Akt activity to suppress cancer cell growth.

Although research has increasingly described the potential effects of citrate on the inhibition of cancer cell growth and sensitizing cancer cells to chemotherapeutic agents [27,28,29,30,31], the spectrum of cellular signaling factors involved in growth inhibition by citrate is poorly understood. In this study, we investigated the underlying mechanisms that involve the inhibition of Akt signaling by citrate linked to the growth arrest of cancer cells.

## 2. Results

### 2.1. Citrate Induces G_2_/M-Phase Arrest in Human Cancer Cells by Stabilizing the Formation of Cyclin B1–p-CDK1 (Thr 161) Complexes

We first examined the effect of citrate on the growth of human pharyngeal squamous carcinoma (PSC) cells using the MTT assay and flow cytometric analysis of propidium iodide (PI) uptake. As shown in Figure 1, citrate did affect PSC cell growth and viability in a dose- and time-dependent manner. Treatment with citrate for 36 h resulted in a decrease in the growth and viability of PSC cells, with half-maximal inhibitory concentration cell viability (IC_50_) values of 10 mM. Thus, we used a citrate concentration of 10 mM for all subsequent experiments.

To investigate whether the reduction in PSC cell growth was due to cell cycle arrest triggered by citrate, its effect on cell cycle progression was examined by flow cytometry of PI-stained cells. With citrate treatment, more cells accumulated in the G_2_/M phase than in vehicle-treated cells. A significant increase in the number of sub-G_1_-phase populations was also observed in citrate-treated cells (Figure 2A). It has been shown that cyclin-dependent kinase 1 (CDK1) is activated by binding to cyclin B1 and phosphorylated at its residue on threonine (Thr) 161, which can then drive G_2_/M phase cell cycle progression [32,33,34]. To examine whether citrate affected the CDK1 activity of treated cells, the level of CDK1 and related proteins regulating the S-G_2_/M phase transition was analyzed. After exposure to citrate, cells showed an increase in the level of cyclin B1 (Figure 2B) and an increase in the level of Thr 161-phosphorylated CDK1 (Figure 3). Coimmunoprecipitation was performed in citrate-treated cell extracts using antibodies specific for CDK1 and cyclin B1 to characterize the effect of citrate on the interaction between CDK1 and cyclin B1. Western blot analysis of the coimmunoprecipitates using an antibody against CDK1 revealed that phospho (p)-CDK1 (Thr 161) formed a complex with cyclin B1 in citrate-treated cells. Reciprocally, cyclin B1–p-pCDK1 (Thr 161) complexes were immunoprecipitated by an antibody against cyclin B1. In contrast, control immunoglobulin G (IgG) antibodies did not immunoprecipitate any specific protein that interacted with cyclin B1 or CDK1 protein (Figure 3), confirming the specificity of the cyclin B1–p-CDK1 (Thr 161) complexes in citrate-treated cells.

Elevating the stability of p21 is known to prevent the activation of cyclin B1–CDK1 and induction of G_2_/M arrest and apoptosis [21,35]. Caspase-3 activity has been shown to be required for the induction of p21 cleavage [36]. To investigate whether the caspase-3-mediated p21 cleavage is involved in citrate-induced G_2_/M arrest, we cotreated the cells with the caspase-3 inhibitor Ac-DEVD-CMK. Citrate-induced p21 cleavage and G_2_/M accumulation were suppressed by Ac-DEVD-CMK (Figure 4A,B). Therefore, we investigated the effect of overexpression of wild-type p21 (wt p21) and caspase-3 cleavage-resistant p21 mutant (p21 (D112N)) on G_2_/M phase arrest. Ectopic expression of wt p21 or p21 (D112N) suppressed citrate-induced cyclin B1 expression and G_2_/M phase arrest (Figure 4C,D). The suppressive effect on the G_2_/M phase arrest of p21 (D112N) was much greater than that of wt p21 (Figure 4D), indicating that proteolytic cleavage of p21 by activated caspase 3 was functionally linked to the induction of G_2_/M.

### 2.2. Suppression of Akt-Mediated Signaling is Required for Citrate-Induced G_2_/M Phase Arrest

To clarify whether G_2_/M phase arrest was induced because of the inhibition of Akt activity by citrate, we examined the effect of the ectopic expression of constitutively active (CA) or dominant-negative (DN) Akt1 proteins on citrate-treated cells. Compared with control vector-transfected cells, forced ectopic expression of HA-CA Akt1 reduced the cyclin B1 level and resulted in a decrease in the number of G_2_/M-phase populations. HA-CA Akt1 expression could reduce citrate-induced p21 cleavage, cyclin B1–p-CDK1 (Thr 161) complex formation, and G_2_/M phase arrest. Overexpression of phosphorylation-deficient HA-DN Akt1 regulated cell growth with significant G_2_/M phase arrest; induced p21 cleavage, cyclin B1 expression, CDK1 (Thr 161) phosphorylation, and cyclin B1–p-CDK1 (Thr 161) complex formation; and enhanced cell cycle arrest in the G_2_/M phase triggered by citrate (Figure 5). These results suggest that an arrest of the cell cycle in the G_2_/M phase requires the suppression of Akt1-mediated signaling by citrate to promote the cleavage of p21, leading to the formation of cyclin B1–p-CDK1 (Thr 161) complexes.

### 2.3. p85α–PTEN Complex-Mediated Inhibition of Akt is Required for Citrate-Induced G_2_/M Phase Arrest

We next examined the levels of p85α, p-p85α (Tyr 508), p110α, PTEN, and p-PTEN (Ser 380/Thr 382/385). As shown in Figure 6A,C, citrate suppressed the phosphorylation of p85α (Tyr 508) and PTEN (Ser 380/Thr 382/385) and reduced the membrane formation of p-p85α (Tyr 508)–p110α complexes, but it did increase the PTEN level. However, citrate treatment did not affect the levels of p85α and p110α (Figure 6A). p85α has been shown to play an important role in stabilizing PTEN by interacting with its dephosphorylated active form at the plasma membrane [37,38]. To examine whether suppressed Akt (Ser 473) phosphorylation is associated with an induction in the formation of p85α–PTEN complexes by citrate, we transfected cells with a PTEN shRNA-expressing plasmid or treated cells with a PTEN inhibitor (SF1670) followed by citrate. The specificity of shRNA for the target PTEN was confirmed by immunoblot analysis (Figure 6A). Knockdown of PTEN in cells enhanced the phosphorylation of Akt (Ser 473) and reduced the level of cyclin B1. Cells expressing PTEN shRNA in the presence of citrate exhibited rescue effects on the suppression of Akt (Ser 473) phosphorylation and the formation of cyclin B1–p-CDK1 (Thr 161) complexes and G_2_/M phase arrest, but PTEN shRNA did not overcome the attenuation of membrane-associated p-p85α (Tyr 508)–p110α complexes by citrate. (Figure 6A–C). The treatment of cells with the PTEN inhibitor (SF1670) alone resulted in an increased level of the phosphorylated form of Akt (Ser 473) and a decreased level of cyclin B1 (Figure 6A). The cotreatment of SF1670 along with citrate restored cyclin B1, Ser 473 phosphorylated Akt, cyclin B1–p-CDK1 (Thr 161) complex formation, and G_2_/M phase to normal levels, similar to the results of treatment with a vehicle control. However, cells treated with SF1670 failed to affect the induction of p85α–PTEN complexes and reduction of p-p85α (Tyr 508)–p110α complexes at the plasma membrane by citrate (Figure 6A–C). These observations indicate that the enhanced complex formation of p85α–PTEN through suppression of p-p85α (Tyr 508)–p110α complex formation is required for citrate-induced Akt inactivation, cyclin B1–p-CDK1 (Thr 161) complex formation, and G_2_/M phase arrest.

## 3. Discussion

Constitutive glycolytic metabolism in cells is correlated with the ability to modulate the expression of antiapoptotic, oncogenic, and tumor-suppressive genes, thereby promoting the development of cancer cells. Targeting the regulators that control the glycolytic pathway is a suggested mechanism for therapeutic cancer strategies [1,39]. In the present work, overexpression of DN Akt1 resulting in the formation of cyclin B1–p-CDK1 (Thr 161) complexes conferred G_2_/M phase arrest. The silencing of PTEN or treatment with the PTEN inhibitor (SF1670) caused an increased level of phosphorylated Akt (Ser 473) and decreased level of cyclin B1. The coimmunoprecipitation assay of membrane fractions of vehicle-treated cells using an antibody specific for p85α or PTEN revealed that p-p85α (Tyr 508) formed a complex with p110α in the plasma membrane but did not interact with PTEN. In the presence of citrate, membrane-associated p85α–PTEN complexes were detected by p85α or PTEN antibody; however, a small amount of p110α could be immunoprecipitated by the anti-p85α antibody. p85α has been shown to bind preferentially to p110α to stabilize p110α and regulate PI3K activity [15]. p110α-free p85α can bind unphosphorylated PTEN and positively regulate its phosphatase activity [15]. Considering these findings and the observed decrease in p-p85α (Tyr 508)–p110α complex formation causing the association of p85α with unphosphorylated PTEN at the plasma membrane after treatment with citrate, it is logical to suggest that the stabilization of p-p85α (Tyr 508)–p110 complexes at the plasma membrane has physiological relevance related to the promotion of Akt activation to allow the progression of the cancer cell proliferation cycle in aerobic glycolytic metabolism.

Phosphorylation of Tyr 508 has been implicated in modulating p85α-mediated conformational change and plasma membrane localization of p110α to act on the lipid kinase activity of PI3K upon platelet-derived growth factor (PDGF) receptor stimulation [40,41,42]. Monomeric p110α is catalytically inactive, requiring a direct interaction with p85α to convert to the active form [43,44]. The ability of p85α to interact with p110α has been proposed to translocate PI3K to the plasma membrane and stimulate PI3K-mediated Akt activation [14]. Active PI3K phosphorylates PIP_2_, converting it to PIP_3_ in an ATP-dependent manner [14,45]. Elevated cytosolic citrate inhibits ATP synthesis via suppressing the activities of pyruvate dehydrogenase and the key glycolytic enzyme phosphofructokinase [8,9,10]. Treatment with citrate did decrease the levels of p-p85α (Tyr 508) and p-Akt (Ser 473). PTEN (Ser 380/Thr 382/385) phosphorylation did not show any change, while PTEN expression was found to be increased in cells with citrate. p-p85α (Tyr 508)–p110α complexes were decreased from the plasma membrane, whereas p85α–PTEN complexes were present in the plasma membrane of citrate-treated cells. Dephosphorylated PTEN represents an active form and requires interaction with p85α at the plasma membrane for the induction of phosphatase activity [15]. Stabilization of PTEN is controlled by p85α interaction, which promotes ubiquitination by E3 ligase WWP2 activity involving ATP-dependent formation [38,46]. The inactivation of PTEN by shRNA or SF1670 had a clear suppressor effect on citrate-induced inhibition of Akt activity and induction of cyclin B1 expression, cyclin B1–p-CDK1 (Thr 161) complexes, and G_2_/M phase arrest, but there was no effect on the levels of p-p85α (Tyr 508) and p-p85α (Tyr 508)–p110α complexes. The induction of p21 cleavage, cyclin B1 expression, CDK1 Thr161 phosphorylation, cyclin B1–p-CDK1 (Thr 161) complex formation, and G_2_/M phase arrest by citrate was reversed by CA Akt1 overexpression. Constitutive Akt inactivation by DN Akt1 ectopic expression could promote p21 cleavage, cyclin B1 expression, CDK1 Thr161 phosphorylation, and cyclin B1–p-CDK1 (Thr 161) complex formation and induce cell cycle arrest during S phase in response to vehicle treatment. These observations confirm and extend previous studies [14,15,26,38] to clearly indicate that plasma membrane-associated p-p85α (Tyr 508)–p110α complexes are key upstream regulators of Akt and PTEN that mediate the formation of cyclin B1–p-CDK1 (Thr 161) complexes and regulate G_2_/M phase progression in cancer cells toward the anaerobic glycolytic process. An inhibitory effect of p21 on the cyclin B1–CDK1 activation has been shown to promote dephosphorylation of CDK1 by Cdc25 [35]. Ectopic expression of p21 can block the phosphorylation of CDK1 (Thr161) by CDK-activating kinase [47]. The observed caspase-3 activation-mediated cyclin B1 expression and G_2_/M arrest induction of citrate was attenuated by ectopic expression of wt p21 or p21 (D112N). Based on these observations, we propose a working model whereby citrate can induce cell cycle arrest in the G_2_/M phase (Figure 7). Citrate suppressing the phosphorylation of p85α (Tyr 508) via the inhibition of ATP synthesis interferes with the interaction of p85α with p110 in the plasma membrane, thereby attenuating the regulatory effect of PI3K-mediated Akt activity on the control of cell cycle G_2_/M phase progression.

The phosphorylation profiles of p85α, rather than phosphorylation of Tyr 508, determine the regulatory function of p85α in heterodimer formation between p85α and p110α and the lipid kinase activity of PI3K. That the phosphorylation of Ser 608, a residue located in an area of the p85α iSH2 domain, confers stable membrane association of p110α to p85α, which determines stabilization of p110α and activity of PI3K lipid kinase, indicates that the phosphorylation of p85α Ser 608 can direct the functional status of PI3K [48,49,50]. Investigation on the underlying mechanisms that involve the suppression of plasma membrane-associated p-p85α (Tyr 508)–p110α complexes by citrate do not rule out the possible involvement of the deregulation of Ser 608 phosphorylation in the process. In the present study, we could not assess the phosphorylation status of Ser 608 in citrate-treated cells because of the lack of an available phospho-specific antibody against p85α Ser 608. However, our findings provide a mechanistic explanation for the recently reported role of an increased PTEN level in the suppression of tumor growth [51].

## 4. Materials and Methods

### 4.1. Cell Culture

The human pharyngeal squamous carcinoma FaDu and Detroit 562 cell lines were obtained as previously described [52,53]. FaDu and Detroit 562 cell lines were cultured in minimum essential medium (MEM) supplemented with 5% fetal bovine serum (FBS). The cell lines were grown in 10-cm tissue culture dishes at 37 °C in a humidified incubator containing 5% CO_2_.

### 4.2. Chemicals, Reagents, and Plasmids

Citrate, dithiothreitol, ethylene glycol tetraactic acid (EGTA), leupeptin, MgCl_2_, 3-(4,5-dimethylthiazol-2-yl)-2,5-diphenyltetrazolium bromide (MTT), *N*-(9,10-dihydro-9,10-dioxo-2-phenanthrenyl)-2,2-dimethyl-propanamide, *N*-(9,10-dioxo-9,10-dihydro-phenanthren-2-yl)-2,2-dimethylpropionamide (SF1670), NaF, Na_3_VO_4_, paraformaldehyde, pepstatin A, phenylmethylsulfonyl fluoride, Tris-HCl, propidium iodide (PI), and Triton X-100 were obtained from Sigma-Aldrich (St. Louis, MO, USA). Citrate was dissolved in and diluted with ethanol and stored at −20 °C as a 100-mM stock. Ethanol and potassium phosphate were purchased from Merck (Darmstadt, Germany). Lipofectamine 2000 was purchased from Thermo Fisher Scientific (New York, NY, USA). DMEM, FBS, glutamine, MEM, penicillin-streptomycin, and trypsin-EDTA were obtained from Gibco BRL (Grand Island, NY, USA). Protein A-agarose beads were purchased from Amersham Biosciences (Piscataway, NJ, USA). The inhibitor of caspase 3 (Ac-DEVD-CMK) was purchased from Calbiochem (San Diego, CA, USA) and dissolved in DMSO. pHA-wt p21, pFLAG-p21 (D112N), pHA-CA Akt1, pHA-DN Akt 1, and pPTEN shRNA vectors were obtained from Addgene (Cambridge, MA, USA).

### 4.3. Antibodies

Antibody against p-CDK1 (Thr 161) was purchased from Abcam (Cambridge, MA, USA). Anti-p85α, anti-Akt, and p-Akt (Ser 473) antibodies were purchased from BD PharMingen. Antibodies against p21, CDK1, CDK2, cyclin A, cyclin B1, cyclin D, cyclin E, and p110α were purchased from Santa Cruz Biotechnology. Anti-pan-cadherin, anti-PTEN, and anti-PTEN (Ser 380/Thr 382/385) antibodies were obtained from Thermo Fisher Scientific (New York, NY, USA). Antibodies against β-actin, hemagglutinin (HA)-epitope tag, and FLAG-epitope tag were obtained from Sigma-Aldrich. Peroxidase-conjugated anti-mouse IgG, anti-goat IgG, and anti-rabbit IgG secondary antibodies were purchased from Jackson ImmunoResearch Laboratory (West Grove, PA, USA).

### 4.4. Cell Viability Assay

The cells were seeded at 3 × 10^4^ cells/well in 24-well tissue culture plates. The cells were grown overnight to ~60% confluence and treated with either ethanol as the vehicle control or citrate for the indicated periods. For a vehicle control, ethanol was diluted in culture medium to the same final concentration of DMSO (0.1%; v/v) as in the medium with citrate. At the end of the incubation, the treated cells were harvested and stained with PI solution (10 μg/mL in phosphate-buffered saline; PBS). The samples were analyzed on a FACSCount flow cytometer (BD Biosciences, Franklin Lakes, NJ, USA). Cell Quest software (BD Biosciences, Franklin Lakes, NJ, USA) was used to analyze the results. PI-negative populations were defined as viable cells.

### 4.5. Cell Proliferation Assay

The cells were seeded at a density of 3 × 10^4^ cells per well into 24-well plates. After 16 h of incubation, the cells were grown to ~60% confluence and treated with either vehicle or citrate at 37 °C for the indicated periods before being harvested. Cell proliferation was determined by the MTT method. The treated cells were washed once with PBS and incubated with 0.5 mg/mL of MTT for 5 h. The resulting formazan precipitate was dissolved in 150 μL of DMSO, and the optical density (OD) of formazan was determined using an ELISA reader (Thermo Labsystems, Multiskan Spectrum, Waltham, MA, USA) at 570 nm.

### 4.6. Measurement of the Cell Cycle by Flow Cytometry

Cells (1 × 10^5^) were trypsinized, washed twice with PBS, and fixed in 80% ethanol. Fixed cells were washed with PBS, incubated with 100 μg/mL of RNase for 30 min at 37 °C, stained with PI (50 μg/mL), and analyzed on a FACSCount flow cytometer. The percentage of cells that had undergone apoptosis was assessed as the ratio of the area of fluorescence that was smaller than the G_0_–G_1_ peak to the total area of fluorescence.

### 4.7. Western Blot Analysis

Treated or transfected cells were lysed in lysis buffer (50 mM Tris-HCl (pH 8.0), 120 mM NaCl, 1 μg/mL of aprotinin, 100 mM Na_3_VO_4_, 50 mM NaF, and 0.5% NP-40). The protein concentration was determined by the Bradford method (Bio-Rad, Hercules, CA, USA). Proteins were separated by electrophoresis on a 10% SDS-PAGE gel and then transferred to a polyvinylidene difluoride membrane (Immobilon-P; Millipore, Bedford, MA, USA). The membranes were blocked overnight with PBS containing 3% skim milk and then incubated with primary antibodies against Akt, p-Akt (Ser 473), cyclin A, cyclin B, cyclin D, cyclin E, CDK1, p-CDK1 (Thr161), CDK2, PTEN, PTEN (Ser 380/Thr 382/385), and p21. The proteins were visualized by staining the membrane with horseradish peroxidase-conjugated goat anti-mouse, goat anti-rabbit, or donkey anti-goat antibodies and the Western blotting luminol reagent. To confirm equal protein loading, β-actin was measured. Densitometric measurements of the band in Western blot analysis were performed using the computed densitometry and ImageQuant 5.2 software (Molecular Dynamics, Sunnyvale, CA, USA).

### 4.8. Plasmid Transfection

Cells (at 70% confluency in a 6- or 12-well plate) were transfected with expression plasmids for HA-CA Akt1, HA-DN Akt1, HA-wt p21, FLAG-p21 (D112N), PTEN shRNA or empty vector using Lipofectamine 2000.

### 4.9. Subcellular Fractionation

Subcellular fractionation was performed according to the protocol of Taha et al. [54]. The treated cells were washed twice with ice-cold PBS and scraped into a detergent-free lysis buffer (10 mM Tris/HCl (pH 7.4), 10 mM NaCl, 0.5 mM MgCl_2_, and EDTA-free protease inhibitor cocktail). The suspension of cells was homogenized using a prechilled 7-mL Dounce homogenizer and then centrifuged at 1200 ×*g* for 5 min at 4 °C. The supernatant was collected and then centrifuged for 5 min at 1200× *g* at 4 °C. The resulting supernatant was further subjected to a 16,000× *g* centrifugation step for 10 min at 4 °C to isolate the heavy membrane pellet. The heavy membrane pellet was reserved as the plasma membrane fraction and lysed in RIPA buffer (1% sodium deoxycholate, 0.1% SDS, 1% Triton X-100, 10 mM Tris-HCl (pH 8.0), and 0.14 M NaCl) for Western blot analysis of the coimmunoprecipitation experiment. The purity of the plasma membrane fraction was confirmed by Western blotting using a specific antibody against the plasma membrane marker cadherin.

### 4.10. Coimmunoprecipitation

Cells (5 × 10^5^) were treated with 10 mM citrate for 36 h and then harvested and lysed in 0.6 mL of cell lysis buffer (50 mM Tris-HCl (pH 8.0), 150 mM NaCl, 1 mM Na_2_EDTA, 100 mM Na_3_VO_4_, 0.5% (v/v) NP-40, 50 mM NaF, 25 mM leupeptin). Whole-cell extracts were incubated with anti-CDK1, anti-cyclin B1, p85α, PTEN, or normal mouse IgG and normal rabbit IgG as a negative control at 4 °C for 2 h. After incubation, protein A-agarose beads were added, and the mixture was incubated with gentle rocking at 4 °C for 2 h. The immunocomplexes were then analyzed by 10% SDS-PAGE and immunoblotted with antibodies against cyclin B1, CDK1, p-CDK1 (Thr 161), p85α, PTEN, and p-PTEN (Ser 380/Thr 382/385).

### 4.11. Statistical Analysis of Data

Statistical calculations of the data were performed using unpaired Student’s *t*-test and ANOVA. Statistical significance between the vehicle control and experimental groups was set at *p* < 0.05.

## 5. Conclusions

Taken together, citrate-elevated p85α interacts with PTEN at the plasma membrane to cause Akt inactivation, leading to the formation of cyclin B1–p-CDK1 (Thr 161) complexes and subsequent induction of G_2_/M phase arrest. This finding not only has potential implications for the current understanding of the molecular mechanism of citrate inhibition of cancer cell growth but also provides a theoretical basis for the further development of novel glycolytic inhibitors.

## Figures and Tables

**Figure 1 ijms-20-02105-f001:**
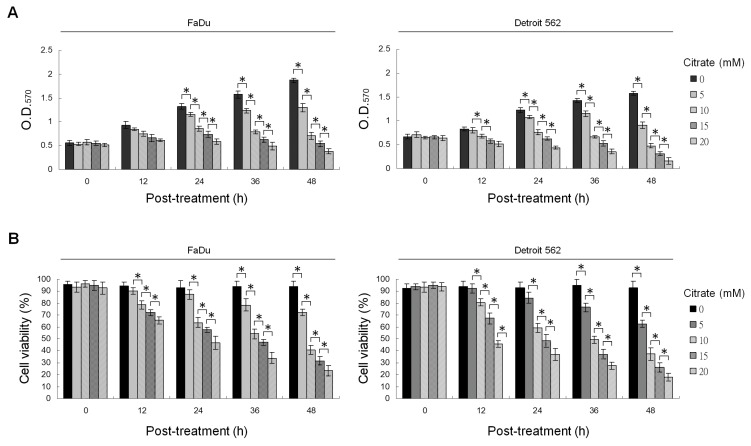
Citrate inhibits the growth of human pharyngeal squamous carcinoma (PSC) cells. (**A**,**B**) Cytotoxic and growth-inhibitory effects of citrate on PSC cells. The cells were treated with vehicle or the indicated concentrations of citrate for the indicated periods of time. Cell viability and proliferation were determined by flow cytometric analysis of propidium iodide (PI) uptake and MTT assay, respectively. The values are presented as the means ± standard error of three independent experiments. * *p* < 0.05: significantly different from vehicle (−)-treated cells.

**Figure 2 ijms-20-02105-f002:**
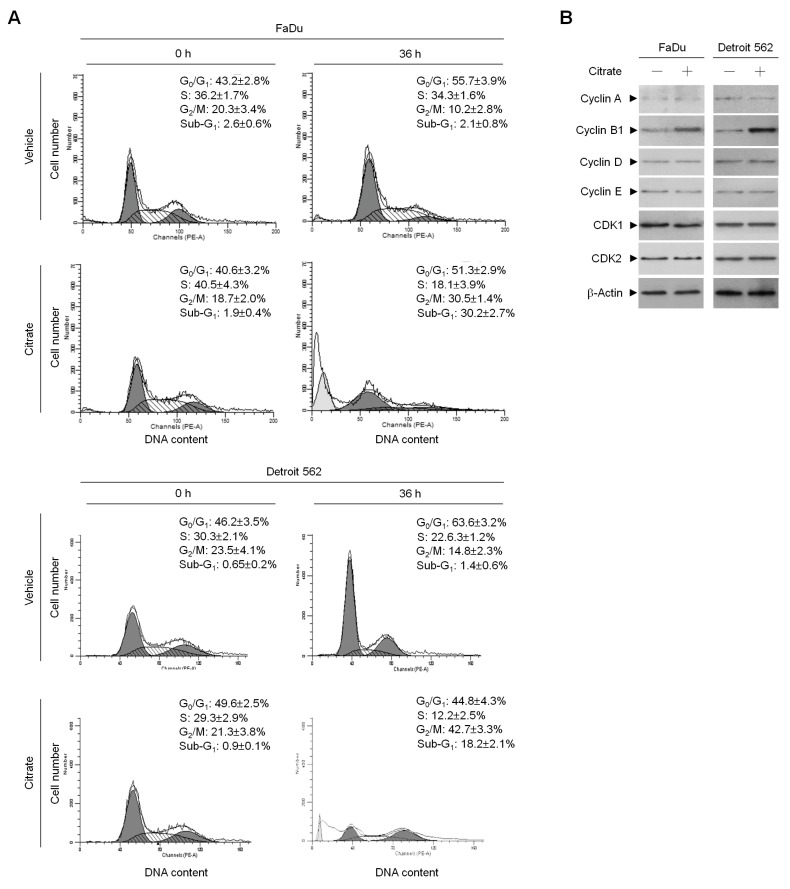
Citrate induces G_2_/M phase arrest of human PSC cells. (**A**,**B**) After treatment of the cells with vehicle (−) or citrate (10 mM) for 36 h, the levels of the indicated proteins in the cell lysates and cell cycle phase were determined using Western blot analysis with specific antibodies and flow cytometric analysis of PI-stained cells, respectively. The values are presented as the means ± standard error of three independent experiments. β-Actin was used as an internal control for sample loading.

**Figure 3 ijms-20-02105-f003:**
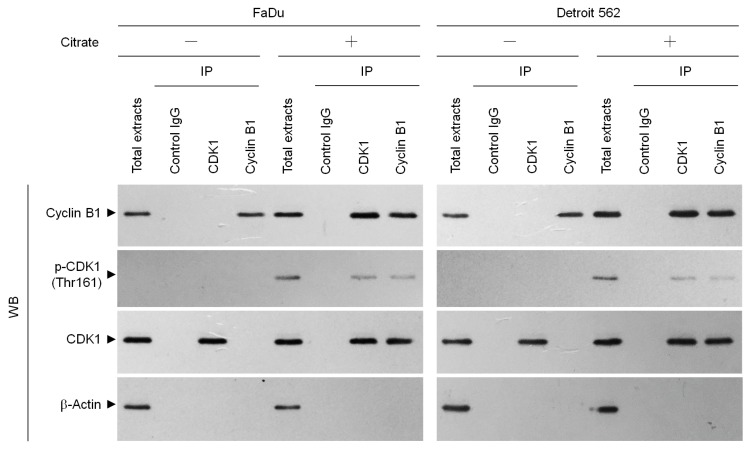
Induction of the formation of cyclin B1–phospho (p)-cyclin-dependent kinase 1 (CDK1) (Thr 161) complexes by citrate in human PSC cells. Cells were treated with vehicle (−) or citrate (10 mM) for 36 h. The antibody used for coimmunoprecipitation is indicated at the top. The proteins from the immunoprecipitated complexes were detected using Western blotting with specific antibodies. Normal immunoglobulin G (IgG) was used as a control for antibody specificity.

**Figure 4 ijms-20-02105-f004:**
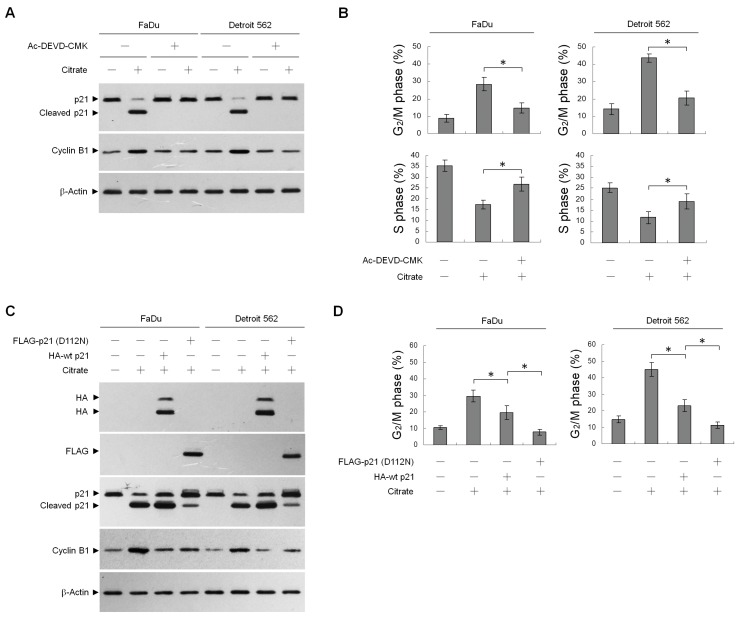
Citrate-induced caspase 3-mediated p21 cleavage is involved in the induction of G_2_/M phase cell cycle arrest. (**A**,**B**) Cells were treated with vehicle (−), citrate (10 mM), Ac-DEVD-CMK (10 μM), or citrate (10 mM) plus Ac-DEVD-CMK (10 μM) for 36 h. (**C**,**D**) At 12 h after transfection with an empty vector, HA-wt p21, or FLAG-p21 (D112N), the cells were treated with either vehicle (−) or citrate (10 mM) for 36 h. The levels of indicated proteins in the cell lysates and cell cycle phase were determined using Western blot analysis with specific antibodies and flow cytometric analysis of PI-stained cells, respectively. β-Actin was used as an internal control for sample loading. The values are presented as the means ± standard error of three independent experiments. * *p* < 0.05: significantly different from empty vector-transfected citrate-treated cells or wt-p21-transfected citrate-treated cells.

**Figure 5 ijms-20-02105-f005:**
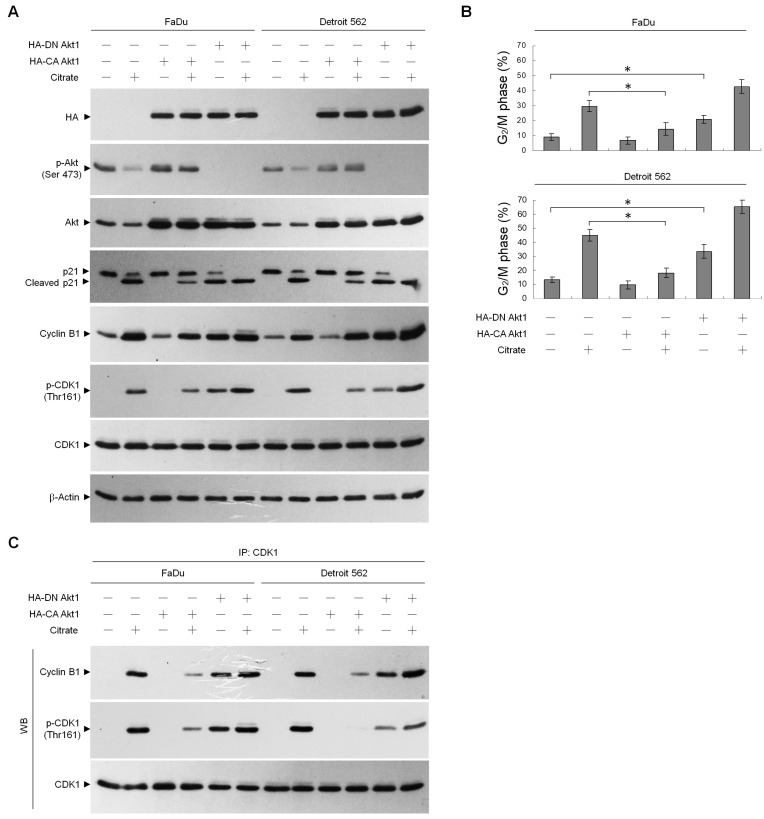
Inhibition of Akt activity by citrate confers the induction of cyclin B1–p-CDK1 (Thr 161) complexes and G_2_/M phase arrest. At 12 h after transfection with an empty vector, HA-CA Akt1, or HA-DN Akt1, the cells were treated with either vehicle (−) or citrate (10 mM) for 36 h. (**A**,**B**) The levels of the indicated proteins in the cell lysates and cell cycle phase were determined using Western blot analysis with specific antibodies and flow cytometric analysis of PI-stained cells, respectively. The values are presented as the means ± standard error of three independent experiments. * *p* < 0.05: significantly different from empty vector-transfected vehicle-treated cells or empty vector-transfected citrate-treated cells. (**C**) The antibody used for coimmunoprecipitation is indicated at the top. The proteins from the immunoprecipitated complexes were detected using Western blotting with specific antibodies. Normal IgG was used as a control for antibody specificity.

**Figure 6 ijms-20-02105-f006:**
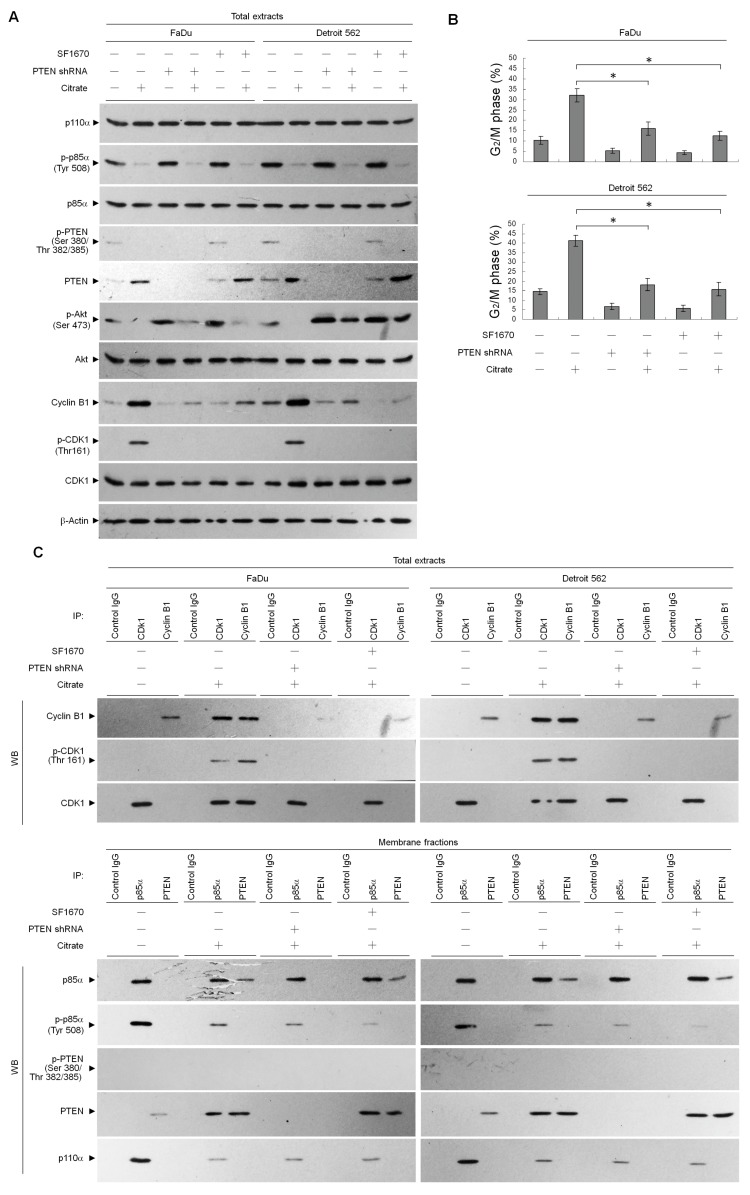
Suppression of p-p85α (Tyr 508)–p110 complex formation by citrate is required for induction of the formation of p85α–phosphatase and tensin homolog deleted from chromosome 10 (PTEN) complexes and G_2_/M phase arrest. At 12 h after transfection with an empty vector or PTEN shRNA, or at 6 h after pretreatment with SF1670 (2 µM), cells were treated with either vehicle (−) or citrate (10 mM) for 36 h. (**A**,**B**) The levels of the indicated proteins in the cell lysates and cell cycle phase were determined using Western blot analysis with specific antibodies and flow cytometric analysis of PI-stained cells, respectively. β-Actin was used as an internal control for sample loading. The values are presented as the means ± standard error of three independent experiments. * *p* < 0.05: significantly different from empty vector-transfected citrate-treated cells. (**C**) Coimmunoprecipitation of CDK1 and cyclin B1 was performed using the total extracts prepared from the cells treated as described above. Coimmunoprecipitation of p85α and PTEN was performed using the membrane fractions prepared from the cells treated as described above. The antibodies used for coimmunoprecipitation are indicated at the top of the figure. The proteins present in the immunoprecipitated complexes were analyzed by Western blot analysis using specific antibodies. Normal IgG was used as a control for antibody specificity.

**Figure 7 ijms-20-02105-f007:**
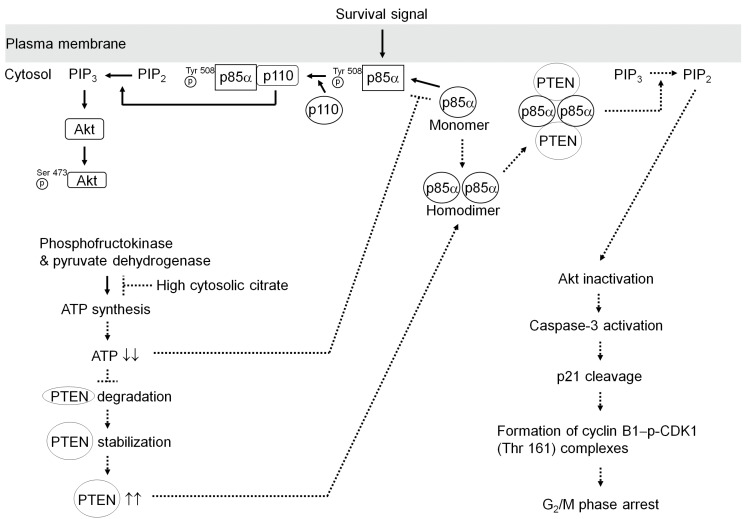
A molecular model for the induction of G_2_/M phase arrest in human PSC cells by citrate. Under the conditions of high cellular citrate uptake, citrate suppresses the activities of pyruvate dehydrogenase and phosphofructokinase, thus inhibiting the production of ATP, attenuating the phosphorylation of p85α (Tyr 508), and interfering with the interaction of p85α with p110α in plasma membrane. Decreased intracellular ATP levels are accompanied by inhibition of the proteasome-mediated degradation of PTEN, which can result in increased levels of PTEN. The resulting p110α-free p85α and unphosphorylated PTEN preferentially form p85α–PTEN complexes in the plasma membrane. Thus, p85α can positively regulate PTEN, resulting in decreased Akt activity and inducing caspase-3-mediated p21 cleavage, cyclin B1–p-CDK1 (Thr 161) complex formation, and an accumulation of cells in G_2_/M phase.

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
