# Peer review of "Citrate-Induced p85α–PTEN Complex Formation Causes G_2_/M Phase Arrest in Human Pharyngeal Squamous Carcinoma Cell Lines"

_ijms, 2019, doi:10.3390/ijms20092105_

Round 1
Reviewer 1 Report
The proposed manuscript “Citrate-Induced p85α-PTEN Complex Formation Causes G2/M Phase Arrest in Human Pharyngeal Squamous Carcinoma Cell Lines” by Kuang-Chen Hung et al. aims at describing the molecular mechanism by which citrate induces the growth arrest of pharyngeal cancer cells. In my opinion, the proposed concept is interesting and globally supported by experimental evidences, but the manuscript is poorly written and needs to be largely reworked, especially the introduction and discussion sections. For these reasons, the manuscript is not acceptable in the present form and a minor revision is suggested:
Globally, the writing of the manuscript is confusing, so a full language revision is strongly suggested. In addition, the authors should explain better what is the rationale of the experiments chosen in order to facilitate the comprehension of the manuscript also to the readers which are not experts of PIK3/Akt pathway. For instance in 2.3 paragraph of the Results section, why did they introduced the analysis of the p85-PTEN complex? The authors briefly could introduce who is p85 and PTEN explained that p85 is a regulatory subunit of PI3K and PTEN is an inhibitor of PI3K/akt pathway activation and so on.
The panel B of figure 6 is not cited in the text.
To complete the figure 6A , the addition of the p21 total and cleaved protein analyses is suggested.
Author Response
Reviewer 1
We thank reviewer for all the constructive points made to improve the quality of the manuscript.
Comment #1 Globally, the writing of the manuscript is confusing, so a full language revision is strongly suggested. In addition, the authors should explain better what is the rationale of the experiments chosen in order to facilitate the comprehension of the manuscript also to the readers which are not experts of PIK3/Akt pathway. For instance in 2.3 paragraph of the Results section, why did they introduced the analysis of the p85-PTEN complex? The authors briefly could introduce who is p85 and PTEN explained that p85 is a regulatory subunit of PI3K and PTEN is an inhibitor of PI3K/akt pathway activation and so on.
[Answer]
1. We have thoroughly checked the manuscript to eradicate grammatical errors and avoided plagiarism using Grammarly: Grammar Checker. The manuscript has been edited by American Journal Experts (Certificate Verification Key#AAAD-4EF5-BA0A-D291-61DE).
2. As we have clearly described in the Introduction section of the original manuscript, (lines 11-18, page 2), Serine/threonine kinase protein kinase B (PKB/Akt)-mediated signaling is critical for proliferation, survival, metabolism, and metastasis of cancer cells [13]), triggered by class IA phosphatidylinositol 3–kinase (PI3K) induction of the generation of phosphatidylinositol-3,4,5-trisphosphate (PIP3) [14]. Although the p85a regulatory subunit is crucial in mediating the activation of PI3K, the ability of p110α-free p85α homodimers to bind and positively regulate the lipid phosphatase activity of phosphatase and tensin homolog deleted from chromosome 10 (PTEN) to modulate Akt-mediated signaling [15]. Akt activity is negatively regulated by PTEN through dephosphorylating the 3-position of PIP3 to phosphatidylinositol-4,5-bisphosphate (PIP2) [16, 17]. We think that a rational analysis of the p85-PTEN complex is therefore not appropriate to describe it again in 2.3. paragraph of the Results section.
Comment #2 The panel B of figure 6 is not cited in the text.
[Answer] As we have clearly stated in 2.3. paragraph of the Results section of the original manuscript (lines 11-21 on page 8), Cells expressing PTEN shRNA in the presence of citrate exhibited rescue effects on the suppression of Akt (Ser 473) phosphorylation and the formation of cyclin B1–p-CDK1 (Thr 161) complexes and G2/M phase arrest, but PTEN shRNA did not overcome the attenuation of membrane-associated p-p85a (Tyr 508)-p110a complexes by citrate. (Figures 6A-C). …… The cotreatment of SF1670 along with citrate restored cyclin B1, Ser 473 phosphorylated Akt, cyclin B1–p-CDK1 (Thr 161) complex formation, and G2/M phase to normal levels, similar to the results of treatment with a vehicle control. However, cells treated with SF1670 failed to affect the induction of p85a-PTEN complexes and reduction of p-p85a (Tyr 508)-p110a complexes at the plasma membrane by citrate (Figures 6A-C).
Comment #3 To complete the figure 6A , the addition of the p21 total and cleaved protein analyses is suggested.
[Answer] We thank the reviewer for the comment. The results present in Figure 6 was aimed to examine whether suppressed Akt (Ser 473) phosphorylation was associated with an induction in the formation of p85a-PTEN complexes by citrate. Using short hairpin RNA against PTEN or PTEN-specific inhibitor SF1670, we have confirm that enhanced complex formation of p85a-PTEN through suppression of p-p85a (Tyr 508)-p110a complex formation is required for citrate-induced Akt inactivation, cyclin B1–p-CDK1 (Thr 161) complex formation, and G2/M phase arrest. The p21 data seems extraneous to focus of the figure 6 study. In fact, it is clear from the data in Figure 5, that an arrest of the cell cycle in the G2/M phase requires the suppression of Akt1-mediated signaling by citrate to promote the cleavage of p21, leading to the formation of cyclin B1–p-CDK1 (Thr 161) complexes (lines 21-23 on page 6). Evidently, Akt1 acts in a regulatory effect to modulate the stability and function of the p21 in FaDu and Detroit 562 cells. Accordingly, p21 cleavage is completely redundant with results shown in Figure 6A.

Reviewer 2 Report
Overall summary
The information that is presented in this manuscript demonstrates that citrate arrests the growth of human pharyngeal squamous carcinoma cells at the G2/M phase by inducing the formation of p85α-PTEN complexes.
Overall, the results support the conclusions that are made by the authors. It is noteworthy to highlight that the figure 7 has been very useful in order to understand the mechanism of action of citrate; so, it is a pity that it had not been referenced in the introduction.
However, as it is noted in detail below, there are a few concerns that should be addressed prior to this paper is considered for publication.
General comments
1. Has the effect of citrate been investigated on normal cells? Is it cytotoxic at concentrations used in this study? Could the effect of citrate be selective?
2. In relation to the results, have you found any difference between the two cell lines used?
3. Why do you think citrate induced also an increase in the number of sub-G1-phase populations? (figure 2)
4. According to the subsection 4.7., the levels of ACL and p-ACL were determined using Western blot analysis. Can you tell me where these results were commented?
5. Why the protein A-agarose beads were not included in the subsection 4.2? Instead of this, the authors have included information about them in lines 11-12, page 14.
6. Legends of the figures must be improved. i) As you know, the figures and their legends must be understandable without reference to the text; ii) sometimes information is duplicated within the legend. For example, the sentence “the levels of indicated proteins in the cell lysates and cell cycle phase were determined using Western blot analysis with specific antibodies and flow cytometric analysis of PI-stained cells, respectively” was included twice in the legend of the figure 4.
7. The references list must be reviewed, concretely: i) you must delete the “,” which appears before the tittle of the articles; ii) the titles of journals must be abbreviated; iii) you must give page spreads instead of abbreviated page numbers.
Other comments
1. Line 18, page 1. The word “an” must be replaced by “as”
2. Lines 14-17, page 2. This sentence is not clear, can you rewrite it?
3. Line 26, page 2. Use present (enhances) instead of past (enhanced)
Author Response
Reviewer 2
We thank reviewer for all the constructive points made to improve the quality of the manuscript.
General comments
Comment #1 Has the effect of citrate been investigated on normal cells? Is it cytotoxic at concentrations used in this study? Could the effect of citrate be selective?
[Answer] A recent in vitro experimental study of citrate uptake in prostate (PC-3M), pancreatic (MiaPaCa-2) and gastric (TMK-1) cancer and in nonneoplastic breast (MCF10A) and prostate (PNT2-C2) cell lines, Mycielska el al. show that cancer cells take up greater amounts of citrate than normal cells via a plasma membrane-specific variant of the mitochondrial citrate transporter. Their experiments show that up to one-third of the total intracellular citrate pool in cancer cells was derived from uptake of extracellular citrate (Cancer Res. 2018 May 15;78(10):2513-2523.). This observation indicates that citrate exhibits selective cytotoxic activity on human cancer cells. Although the present study was not investigated the effect of citrate on human normal cells, according to Mycielska el al. results, we have therefore suggest that citrate used in the treatment of concentration displays low cytotoxicity toward human normal cells.
Comment #2 In relation to the results, have you found any difference between the two cell lines used?
[Answer] The present experimental data from gene manipulation and molecular biochemical assays showed no significant differences in the inductive effect of citrate on p85a-PTEN complex-mediated Akt inactivation, caspase 3-mediated p21 cleavage, cyclin B1–p-CDK1 (Thr 161) complex formation, and G2/M arrest in both FaDu and Detroit 562 cells.
Comment #3 Why do you think citrate induced also an increase in the number of sub-G1-phase populations? (figure 2)
[Answer] It is just an indication that a very late apoptotic stage has been induced in citrate-treated PSC cells.
Comment #4 According to the subsection 4.7., the levels of ACL and p-ACL were determined using Western blot analysis. Can you tell me where these results were commented?
[Answer] When I check the manuscript, I find it wrong to make a mistake. I apologize for my error. Page 13 (lines 28-31), the original description for “The membranes were blocked overnight with PBS containing 3% skim milk and then were incubated with primary antibodies against Akt, p-Akt (Ser 473), cyclin A, cyclin B, cyclin D, cyclin E, CDK1, p-CDK1 (Thr161), CDK2, ACL, p-ACL (Ser 454), and p21.” was changed to “The membranes were blocked overnight with PBS containing 3% skim milk and then were incubated with primary antibodies against Akt, p-Akt (Ser 473), cyclin A, cyclin B, cyclin D, cyclin E, CDK1, p-CDK1 (Thr161), CDK2, PTEN, PTEN (Ser 380/Thr 382/385), and p21.”.
Comment #5 Why the protein A-agarose beads were not included in the subsection 4.2? Instead of this, the authors have included information about them in lines 11-12, page 14.
[Answer] Description of the source of protein A-agarose beads was added in the 4.2. paragraph of the 4. Materials and Methods section on page 12 (lines 31-32) and shown below. Protein A-agarose beads were purchased from Amersham Biosciences (Piscataway, NJ, USA). Page 14 (lines 11-12), the original description for “After incubation, protein A-agarose beads (Amersham Biosciences, Piscataway, NJ, USA) were added, and the mixture was incubated with gentle rocking at 4°C for 2 h.” was changed to “After incubation, protein A-agarose beads were added, and the mixture was incubated with gentle rocking at 4°C for 2 h.”.
Comment #6 Legends of the figures must be improved. i) As you know, the figures and their legends must be understandable without reference to the text; ii) sometimes information is duplicated within the legend. For example, the sentence “the levels of indicated proteins in the cell lysates and cell cycle phase were determined using Western blot analysis with specific antibodies and flow cytometric analysis of PI-stained cells, respectively” was included twice in the legend of the figure 4.
[Answer] Per the reviewer’s comment, legend for figure 4 has been rearranged and clarified by providing the written information as shown below:
1. Page 6 (lines 3-10), the original description for “Figure 4. Citrate-induced caspase 3-mediated p21 cleavage is involved in the induction of G2/M phase cell cycle arrest. (A and B) Cells were treated with vehicle (-), citrate (10 mM), Ac-DEVD-CMK (10 mM), or citrate (10 mM) plus Ac-DEVD-CMK (10 mM) for 36 h, and the levels of the indicated proteins in the cell lysates and cell cycle phase were determined using Western blot analysis with specific antibodies and flow cytometric analysis of PI-stained cells, respectively. The values are presented as the means ± the standard error of three independent experiments. *P < 0.05: significantly different from citrate-treated cells. (C and D) At 12 h after transfection with an empty vector, HA-wt p21, or FLAG-p21 (D112N), the cells were treated with either vehicle (-) or citrate (10 mM) for 36 h, and the levels of indicated proteins in the cell lysates and cell cycle phase were determined using Western blot analysis with specific antibodies and flow cytometric analysis of PI-stained cells, respectively. b-Actin was used as an internal control for sample loading. The values are presented as the means ± standard error of three independent experiments. *P < 0.05: significantly different from empty vector-transfected citrate-treated cells or wt-p21-transfected citrate-treated cells.” was changed to “Figure 4. Citrate-induced caspase 3-mediated p21 cleavage is involved in the induction of G2/M phase cell cycle arrest. (A and B) Cells were treated with vehicle (-), citrate (10 mM), Ac-DEVD-CMK (10 mM), or citrate (10 mM) plus Ac-DEVD-CMK (10 mM) for 36 h. (C and D) At 12 h after transfection with an empty vector, HA-wt p21, or FLAG-p21 (D112N), the cells were treated with either vehicle (-) or citrate (10 mM) for 36 h. The levels of indicated proteins in the cell lysates and cell cycle phase were determined using Western blot analysis with specific antibodies and flow cytometric analysis of PI-stained cells, respectively. b-Actin was used as an internal control for sample loading. The values are presented as the means ± standard error of three independent experiments. *P < 0.05: significantly different from empty vector-transfected citrate-treated cells or wt-p21-transfected citrate-treated cells.”.
Comment #7 The references list must be reviewed, concretely: i) you must delete the “,” which appears before the tittle of the articles; ii) the titles of journals must be abbreviated; iii) you must give page spreads instead of abbreviated page numbers.
[Answer] We thank the reviewer for the comment. The references of manuscript were written in the EndNote pre-formatted “International Journal of Molecular Sciences” template, and it’s then prepared and formatted strictly according to the Journal’s guidelines for authors.
Other comments
Comment #1 Line 18, page 1. The word “an” must be replaced by “as”
[Answer] As suggested, the original description for “Abstract: Citrate is a key intermediate of the tricarboxylic acid cycle and acts an allosteric signal to regulate the production of cellular ATP.” was changed to “Abstract: Citrate is a key intermediate of the tricarboxylic acid cycle and acts as allosteric signal to regulate the production of cellular ATP.” (lines 18-19, page 1).
Comment #2 Lines 14-17, page 2. This sentence is not clear, can you rewrite it?
[Answer] Per the reviewer’s comment, the original descriptions for “Serine/threonine kinase protein kinase B (PKB/Akt)-mediated signaling is critical for proliferation, survival, metabolism, and metastasis of cancer cells [13]), triggered by class IA phosphatidylinositol 3–kinase (PI3K) induction of the generation of phosphatidylinositol-3,4,5-trisphosphate (PIP3) from phosphatidylinositol-4,5-bisphosphate (PIP2) [14]. Although the p85a regulatory subunit is crucial in mediating the activation of PI3K, the ability of p110α-free p85α homodimers to bind and positively regulate the lipid phosphatase activity of phosphatase and tensin homolog deleted from chromosome 10 (PTEN) to modulate Akt-mediated signaling [15]. Akt activity is negatively regulated by PTEN through dephosphorylating the 3-position of PIP3 to PIP2 [16, 17].” was changed to “Serine/threonine kinase protein kinase B (PKB/Akt)-mediated signaling is critical for proliferation, survival, metabolism, and metastasis of cancer cells [13]), triggered by class IA phosphatidylinositol 3–kinase (PI3K) induction of the generation of phosphatidylinositol-3,4,5-trisphosphate (PIP3) [14]. Although the p85a regulatory subunit is crucial in mediating the activation of PI3K, the ability of p110α-free p85α homodimers to bind and positively regulate the lipid phosphatase activity of phosphatase and tensin homolog deleted from chromosome 10 (PTEN) to modulate Akt-mediated signaling [15]. Akt activity is negatively regulated by PTEN through dephosphorylating the 3-position of PIP3 to phosphatidylinositol-4,5-bisphosphate [16, 17].” (lines 11-18, page 2).
Comment #3 Line 26, page 2. Use present (enhances) instead of past (enhanced)
[Answer] Per the reviewer’s comment, the original descriptions for “Previous studies have indicated that Akt enhanced the efflux of citrate from the mitochondria to the cytosol in the form of acetyl-CoA by phosphorylating and activating ACL [23-25].” was changed to “Previous studies have indicated that Akt enhances the efflux of citrate from the mitochondria to the cytosol in the form of acetyl-CoA by phosphorylating and activating ACL [23-25].” (lines 26-27, page 2).
